# The Improvement and Application of the Geoelectrochemical Exploration Method

**Ming Kang [1,*], Huanzhao Guo [1], Wende Zhu [2], Xianrong Luo [3] and Jianwen Yang [4]**

[1] School of Earth Science and Resources, Chang'an University, 126 Yanta Road, Xi'an 710054, China
[2] Geologic Team 212 of Shanxi Provincial Geological Prospecting Bureau, Changzhi 046000, China
[3] College of Earth Sciences, Guilin University of Technology, Guilin 541006, China
[4] Department of Earth and Environmental Sciences, University of Windsor, 401 Sunset Avenue, Windsor, ON N9B 3P4, Canada
[*] Correspondence: kangmin@chd.edu.cn

**Abstract:** The anionic and cationic species of elements from deeply buried deposits migrate to the near surface driven by various geological forces. The geoelectrochemical exploration method (GEM), derived from CHIM, consists of the application of an electric field to collect these active ions at the designated electrode. Prospecting effects have been investigated by researchers since the coming up of CHIM. However, the cumbersome technical equipment, complex techniques and low production efficiency have restricted its potential application in field geological survey. This paper presents the newly developed CHIM that is electrified by a low voltage dipole. The improved technique allows both anionic and cationic species of elements to be extracted simultaneously in an anode and in a cathode. Compared with the conventional CHIM method, the innovative techniques called dipole geoelectrochemical method are characterized by simple instrumentation, low cost and easy operation in field, and in particular enables simultaneous extraction of anionic and cationic species of elements, from which more information can be derived with higher extraction efficiency. The dipole geoelectrochemical method was proposed and applied in the experiments of the Yingezhuang gold ore from Zhaoyuan, Shandong Province, the 210 gold ore from Jinwozi, Xinjiang Province, and the Daiyinzhang gold polymetallic deposit from Wutaishan, Shanxi Province. There are clearly anomalies above the gold ore body, indicating the effectiveness and feasibility of the improved dipole geoelectrochemical method in both scientific research and mineral exploration. The results of anode extraction in several mining areas have shown good results, indicating that gold may be mainly negatively charged. In fact, many metal nanoparticles, clay minerals, or complexes of metal ions are negatively charged, so they migrate to the anode electrode and enrich.

**Keywords:** geoelectrochemical method; dipole; low voltage; technique improvement; concealed deposit





## 1. Introduction

The geoelectrochemical exploration method (GEM), derived from CHIM (Chastichnoe Izvlechennye Metallov), was invented by Leningrad researchers in the late 1960s to early 1970s, and the method refers to partial extraction of metals. The systematic theory and field techniques, together with some practical results, were first published by Ryss and Goldberg (1973) [1]. The laboratory results upon which the method is based, some additional field conditions, equipment parameters, and speed of coverage of the method were described in several other papers [1–3].

In the 1970s and 1980s, the CHIM method was extensively applied in Russia in exploration for base and precious metals, W, U, Be, and oil and gas [4]. In the early 1980s, experimental research was carried out in China [5,6], and in the late 1980s, geoelectrochemical experimentation was started in India [7]. In the 1990s, this method was applied on a trial basis in the USA [8] and Canada [9], and then widely applied to search for concealed ore deposits [10,11]. Since the 2000s, a large amount of research on the halo-forming

mechanism of this method has been performed, and great progress has been made for its technique improvement [12–15]. Excellent results of searching hidden ore deposits of Cu, Pb, Zn, Au, Ag, Sn, As and Sb have been obtained with this method [16]. Meanwhile, application studies have been carried out for different stages of mineral exploration, with great success [17,18].

The CHIM method has been improved since it was put forward. MDI (Method of Diffusion Extraction) was first proposed for exploration of buried ore bodies by Milkov et al. (1981) [19], "Dipole" CHIM was proposed by Levitski (1993) [20], and NEOCHIM was set up by Leinz and Hoover (1993) [21]. More recently, the adsorption–electric extraction method was developed by Fei (1992) [22], Liu et al. (1997) [14], Luo (1994) [23] and Tan and Cai (2000) [15]. However, there are still many disadvantages as follows:

(1) Formerly, researchers [1–11] intended to extract metal ions, under the influence of electric current, from a buried deposit through hundreds of meters of overburden into the element-collectors (ECs) on the surface. To achieve this, a large power generator is required, and each measurement point is connected with the generator by long wires. Procedures are complex, and the operation is inconvenient in field, e.g., the power is easily broken off by accidental factors, such as cars passing by, animal actions, and people walking, etc.

(2) Element-collectors (ECs) filled with solution may leak, to a greater or lesser degree, so that the remaining amount of the solution is very different when the work is finished, which may have an influence on identifying anomalies.

(3) Element-collectors (ECs) of the adsorption–electric extraction method have some advantages over the liquid-type ECs. Previous work, however, only employed a single cathode to extract ions that are in positively charged forms of elements. Levitski (1993) [20] proposed the "Dipole" CHIM method, which enables a simultaneous extraction of anionic and cationic species of elements, and the ECs have been improved greatly as well. However, it is difficult to use in field owing to its weight, complicated operation procedures, high cost and relatively low production rate.

(4) The improvement of method has received only minimal attention in the English-language literature. Kang et al. (2003) [12] and Kang et al. (2006) [13] gave a brief overview of the development and application of the improved method in Chinese, with an English abstract.

On the improvement of the CHIM method, the authors have carried out some exploration in China and made some progress [12,13], and other researchers [24] also conducted related research. In order to carry out the research and further improve the technology of this method on a global scale, the authors systematically summarized the previous research results and wrote this paper.

Firstly, the development history of geoelectrochemical exploration method (GEM) was introduced in this paper. Secondly, the halo-forming mechanism was illustrated on mineral exploration. Then, the basic and the improved geoelectrochemical methods including equipment and sampling were described, respectively. Finally, three case studies on concealed gold deposits in China were illustrated to show the applications of the improved geoelectrochemical method developed.

## 2. Halo-Forming Mechanism

Deeply concealed ore deposits are dissolved in many forms, such as electrochemical dissolution. Metal anionic and cationic species of elements from concealed deposits migrate to near the surface driven by various geological forces, and they are enriched therein [21,25–28]. Ions with a negative charge go to the anode and positive to the cathode under the influence of an artificial electric field. The man-made electric field can activate and change the forms of occurrence of elements in the soil. Firstly, it can bring about decomposition of a great number of complex anions and other stable or sub-stable form of elements; secondly, it can make anions and cations move to the extraction electrodes, and hence accelerate ionic movement [1,17]. The metal ions of electromobile forms are extracted

in either anodes or cathodes under the influence of a man-made electric field, which are called GEM anomalies (Figure 1). There are electrically active fine-grained clay mineral particles in the soil, and charged ions or electrically active ultrafine-grained clay mineral particles can migrate to the designated electrode and be adsorbed in the foam sample under the action of an external electric field [29]. GEM ionic halos are in mobile forms, and they are dynamically related to concealed deposits [7,22,26]. The element composition of the halos is normally correlated with that of the ores, and the halos occur typically directly over the deposits [30]. Therefore, GEM ionic halos can be used to search for concealed deposits.

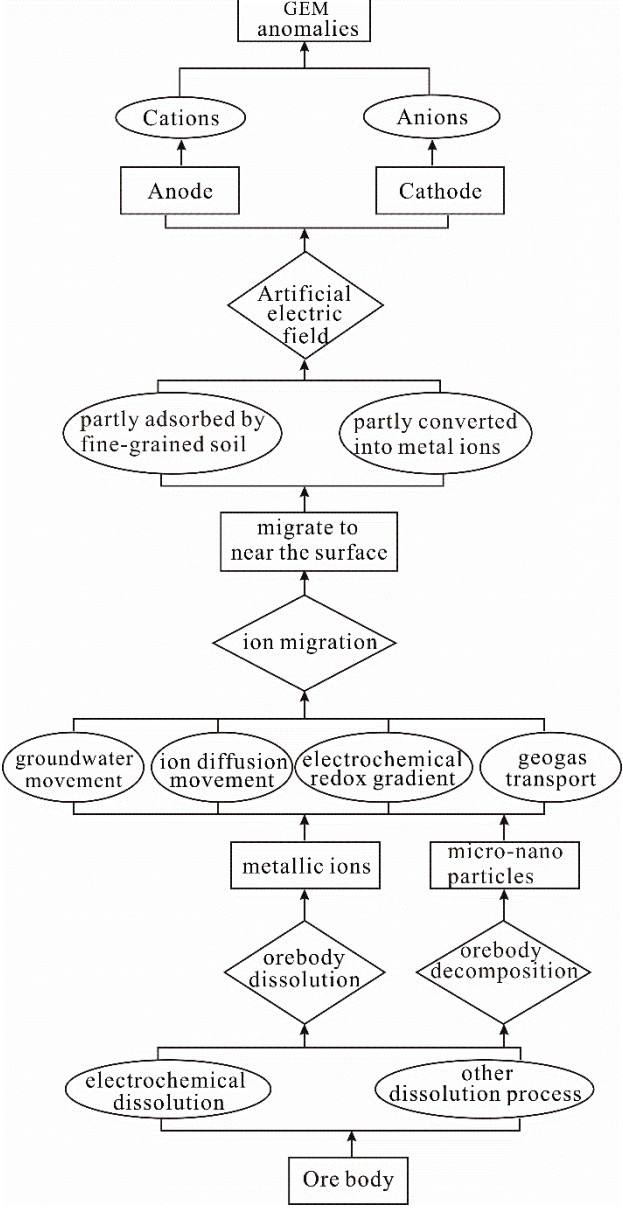

**Figure 1.** The formation of geoelectrochemical ionic halos.

## 3. Materials and Methods

The GEM is a prospecting technique that combines geochemical and geophysical exploration. That is, ions or charged complexes of the electromobile forms of elements in soils and rocks are extracted, under the influence of an artificial electric field, into the specially designed element-collectors (ECs). The ECs are analyzed by ICP-MS for indicator elements related to ore deposits.

### 3.1. Basic GEM-CHIM

The ECs are embedded in the surface sediments along a profile to be explored and are connected to a DC current source as either anodes or cathodes. The collectors comprise cylindrical polyethylene vessels with a semi-permeable diaphragm as a base, filled with specific electrolyte in which a solid electrode is dipped. A common auxiliary electrode is positioned at "infinity" and is represented by a metal or graphite bar(s). The extraction of electromobile forms of occurrence of metals is made with an applied current of 100–200 mA, usually for a duration of 10 or 20 h [7,30–33]. However, the basic CHIM field procedure is complicated and of high cost. As such, some improvements have been made recently, as introduced in this paper.

### 3.2. Improved GEM—"Dipole CHIM" Electrified by a Low Voltage Dipole (Abbreviated as DL-CHIM)

#### 3.2.1. Theoretical Basis

Both cationic and anionic species may bear useful information, and many metals may form anionic complexes, especially in the presence of chlorides, particularly in surface soils. Such complexes include $[CuI]^-$, $[CuCl_2]^-$, $[Cd(NH_3)_2 Cl_4]^{2-}$, $[HgCl_4]^{2-}$, etc. [34]. Recognizing that anionic as well as cationic species may provide useful information [33], the "Dipole CHIM" technique electrified by a low voltage dipole was then proposed.

#### 3.2.2. Method Setting and Field Procedures

The element-collectors comprise a graphite bar wrapped foam electrode pairs connected by a 9-volt alkaline battery (Figure 2). The electrodes were embedded in a rectangular sampling area with a length of 40 cm, a width of 20 cm and a depth of 30 cm, and were placed parallel at the bottom of the sampling area at an interval of 30 cm, and then they were covered with soil while watering and pouring a dilute nitric acid solution of 15% $HNO_3$. Generally, water is in the anode area for one liter, and dilute nitric acid solution is in the cathode area for one liter. These were left for about 24 h, after which time the electrodes were exhumed (depending on the DC power supply and local geological controls). The electrodes need cleaning before using again.

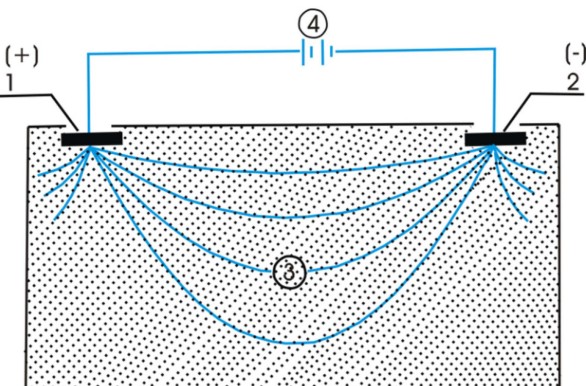

**Figure 2.** Simplified profile of the DL-CHIM. 1—Anion collector; 2—Cation collector; 3—Current flow lines; 4—Disposable DC power supply.

Soil samples were collected at each sampling point to compare the application effects between the DL-CHIM method with the conventional geochemical soil survey (CGS).

### 3.3. Sample Testing

Acid digestion in pretreating foam samples was made, and the contents of Au were determined by Inductively Coupled Plasma-Mass Spectrometry (ICP-MS), and the analytical work was carried out at Institute of Geophysical and Geochemical Exploration, Langfang, China and Guilin Research Institute of Geology for Mineral Resources, Guilin.

## 4. Results and Discussion

### 4.1. Case Study on Yingezhuang Gold Deposit

The DL-CHIM method was applied at Yingezhuang gold deposit from Shandong, 210 gold deposit from Xinjiang, and Daiyinzhang gold polymetallic deposit from Shanxi, China. The Yingezhuang gold deposit is located in the East of China, whose ore body is situated at a depth of between 100 m and 600 m. The average grade of the ore body is 4.03 g/t with a thickness ranging from 2 m to 10 m [13]. Figure 3a,b show the results obtained using the DL-CHIM method. It can be seen from Figure 3 that the DL-CHIM method identifies distinct anomalies of Au over the gold ore bodies, and that cationic species anomalies of Au have highest values (41.05 ppb) above the buried ore bodies, and anion anomalies of Au have high intensity and good continuity, which corresponds closely to ore body. Anionic species anomalies and cationic species anomalies of Au are much wider and have better continuity (sample numbers from 18 to 54), which can give anomalous responses from deep-seated mineralization. Anionic and cationic species anomalies of Au enable the identification of the position of the deeply buried ore bodies. The maximum anomalous value in cationic species anomalies of Au clearly corresponds to the gold-rich ore body of Yingezhuang gold deposit. In the thick transported covered terrains (sample numbers from 18 to 36), and anion anomalies of Au can show better results for concealed gold deposits in depth because gold may mainly exist as micro–nano particles in surface soils, which are negatively charged and can migrate to the designated electrode and be adsorbed in the foam sample under the action of an artificial electric field. The analytical results of this field work are tabulated in Table 1.

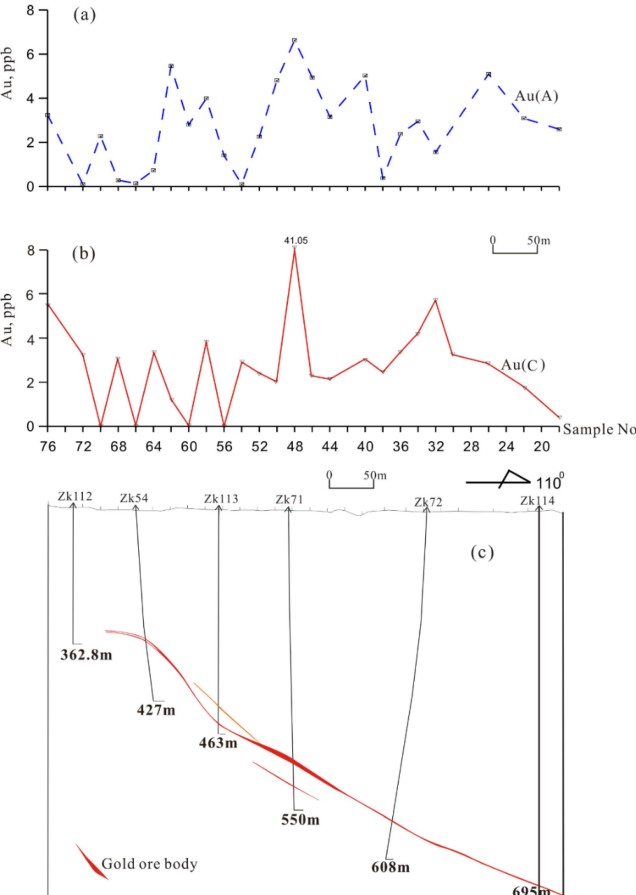

**Figure 3.** Results obtained by employing the DL-CHIM method over Yingezhuang gold deposit from Shandong, China. (**a**) Anion anomalies of Au (Anode extraction); (**b**) Cation anomalies of Au (Cathode extraction); (**c**) geological base map [13].

**Table 1.** Analytical data of the samples obtained using the DL-CHIM at the Yingezhuang gold deposit.

| Sample No. | Anion-Collectors | Cation-Collectors |
|:---:|:---:|:---:|
| | Au (ppb) | Au (ppb) |
| 18 | 2.59 | 0.39 |
| 22 | 3.09 | 1.81 |
| 26 | 5.10 | 2.84 |
| 30 | | 3.28 |
| 32 | 1.56 | 5.68 |
| 34 | 2.95 | 4.23 |
| 36 | 2.38 | 3.36 |
| 38 | 0.38 | 2.49 |
| 40 | 5.03 | 3.05 |
| 44 | 3.15 | 2.18 |
| 46 | 4.94 | 2.31 |
| 48 | 6.63 | 41.05 |
| 50 | 4.81 | 2.06 |
| 52 | 2.25 | 2.43 |
| 54 | 0.10 | 2.88 |
| 56 | 1.40 | 0.10 |
| 58 | 3.99 | 3.80 |
| 60 | 2.80 | 0.10 |
| 62 | 5.46 | 1.20 |
| 64 | 0.73 | 3.30 |
| 66 | 0.14 | 0.10 |
| 68 | 0.28 | 3.08 |
| 70 | 2.28 | 0.10 |
| 72 | 0.1 | 3.26 |
| 76 | 3.24 | 5.60 |
| 80 | 1.64 | 2.14 |

*4.2. Case Study on 210 Gold Deposit*

The 210 gold deposit from Xinjiang is located in the Northwest of China, where arid residual regolith is about 10 m thick. The ore bodies are hosted in a mylonite belt at depths between 20 m and 60 m, whose grade varies between 3 g/t and 10 g/t with an average of 4.11g/t [13]. Figure 4 shows the results obtained using the DL-CHIM method along line I at the 210 gold deposit. It can be seen from Figure 4 that there are obvious anionic species anomalies of Au above the ore body, and the anomalies of Au have a certain continuity, indicating that gold may mainly exist as micro–nano particles, clay minerals, or complexes of metal ions in surface soils, which is negatively charged. Figure 4 indicates that the cation anomaly of Au only shows a single anomalous point. Thus, it suggests that the DL-CHIM method is a great improvement over the previous monopole extraction. The analytical results of this field work are tabulated in Table 2.

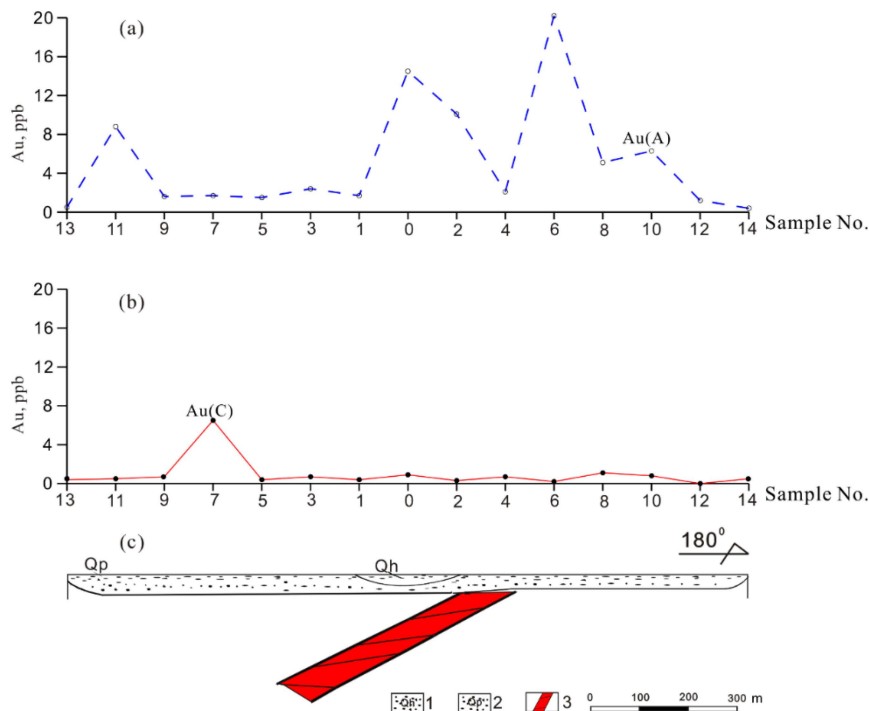

**Figure 4.** Results obtained by employing the DL-CHIM method over 210 gold deposit from Xinjiang, China. 1—Quaternary System Holocene; 2—Quaternary System Pleistocene; 3—Gold ore body; (**a**) Anion anomalies of Au (Anode extraction); (**b**) Cation anomalies of Au (Cathode extraction); (**c**) geological base map [13].

**Table 2.** Analytical results of the samples obtained using DL-CHIM method at the 210 gold deposit.

| Sample No. | Anion-Collectors | Cation-Collectors |
| --- | --- | --- |
| | Au (ppb) | Au (ppb) |
| 0 | 0.9 | 14.5 |
| 1 | 0.4 | 1.7 |
| 2 | 0.3 | 10.1 |
| 3 | 0.7 | 2.4 |
| 4 | 0.7 | 2.1 |
| 5 | 0.4 | 1.5 |
| 6 | 0.2 | 20.2 |
| 7 | 6.5 | 1.7 |
| 8 | 1.1 | 5.1 |
| 9 | 0.7 | 1.6 |
| 10 | 0.8 | 6.3 |
| 11 | 0.5 | 8.8 |
| 12 | 0.0 | 1.2 |
| 13 | 0.4 | 0.5 |
| 14 | 0.5 | 0.4 |

### 4.3. Case Study on Daiyinzhang Gold Deposit

The Daiyinzhang gold polymetallic deposit is located in the midwestern section of Wutaishan area from Shanxi, in the middle-high mountainous area. The terrain in the area is high in the east and low in the west, with developed valleys and severe topography, where exposed strata mainly consist of the chlorite schist and sericite schist of the Baizhiyan Formation of the Neoarchean Wutai Group. The alteration phenomena of pyrite mineralization, silicification, tourmaline, sericitization, and carbonation are obvious. The intrusive

rocks are mainly dominated by Wutai plagioclase granite and Lvliang metamorphic diabase [35,36]. The ore body, with strike length 450 m, has a burial depth of 0 m to 558 m and a thickness of 0.53 m to 2.44 m; its Au average grade is 2.97 g/t. It can be seen from Figure 5 that the cation anomalies of Au mainly occur at measurement points from 11 to 18, and the anomalous values are in the range from 6.21 ppb to 11.21 ppb, with an average intensity of 8.83 ppb, which basically corresponds to the ore bodies near the surface; in addition, the extreme values of the anomalies are relatively continuous, which is basically consistent with the distribution of gold ore bodies. The anion anomalies of Au occur at measurement points from 5 to 11, and the anomalous values are in the range from 4.65 ppb to 16.98 ppb, with an average intensity of 9.47 ppb, corresponding to the buried ore bodies. In the geochemical soil survey, the soil anomalies of Au occur at measurement points from 15 to 19, and the anomalous values are in the range from 4.86 ppb to 481.62 ppb, with an average intensity of 121.02 ppb, which has some displacement with the ore bodies near the surface; for buried ore bodies, the soil anomalies of Au are weak, basically showing the background characteristics. In general, in the thick transported covered terrains (measurement points from 1 to 10), the geochemical soil survey shows only background characteristics; however, DL-CHIM method shows obvious anion and cation anomalies of Au at measurement points for 5, 6 and 8. The analytical results of this field work are tabulated in Table 3.

**Table 3.** Analytical results of the samples obtained using DL-CHIM method at the Daiyinzhang gold polymetallic deposit.

| Sample No. | Anion-Collectors | Cation-Collectors | Geochemical Soil Survey |
|:---:|:---:|:---:|:---:|
| | Au (ppb) | Au (ppb) | Au (ppb) |
| 1 | 5.34 | 13.57 | 2.76 |
| 2 | 5.25 | 5.49 | 1.58 |
| 3 | 7.65 | 4.78 | 2.42 |
| 4 | 5.74 | 5.79 | 1.57 |
| 5 | 16.98 | 8.28 | 1.38 |
| 6 | 4.65 | 10.37 | 2.13 |
| 7 | 5.14 | 4.72 | 2.81 |
| 8 | 9.70 | 4.68 | 1.29 |
| 9 | 4.93 | 6.09 | 1.28 |
| 10 | 13.14 | 4.63 | 1.48 |
| 11 | 11.75 | 11.21 | 4.55 |
| 12 | 6.34 | 9.07 | 2.90 |
| 13 | 5.73 | 9.87 | 9.28 |
| 14 | 5.31 | 9.01 | 8.43 |
| 15 | 5.07 | 7.60 | 38.55 |
| 16 | 4.78 | 9.79 | 4.86 |
| 17 | 7.57 | 7.86 | 481.62 |
| 18 | 4.81 | 6.21 | 53.16 |
| 19 | 6.67 | 8.01 | 26.90 |
| 20 | 4.08 | 7.34 | 1.94 |

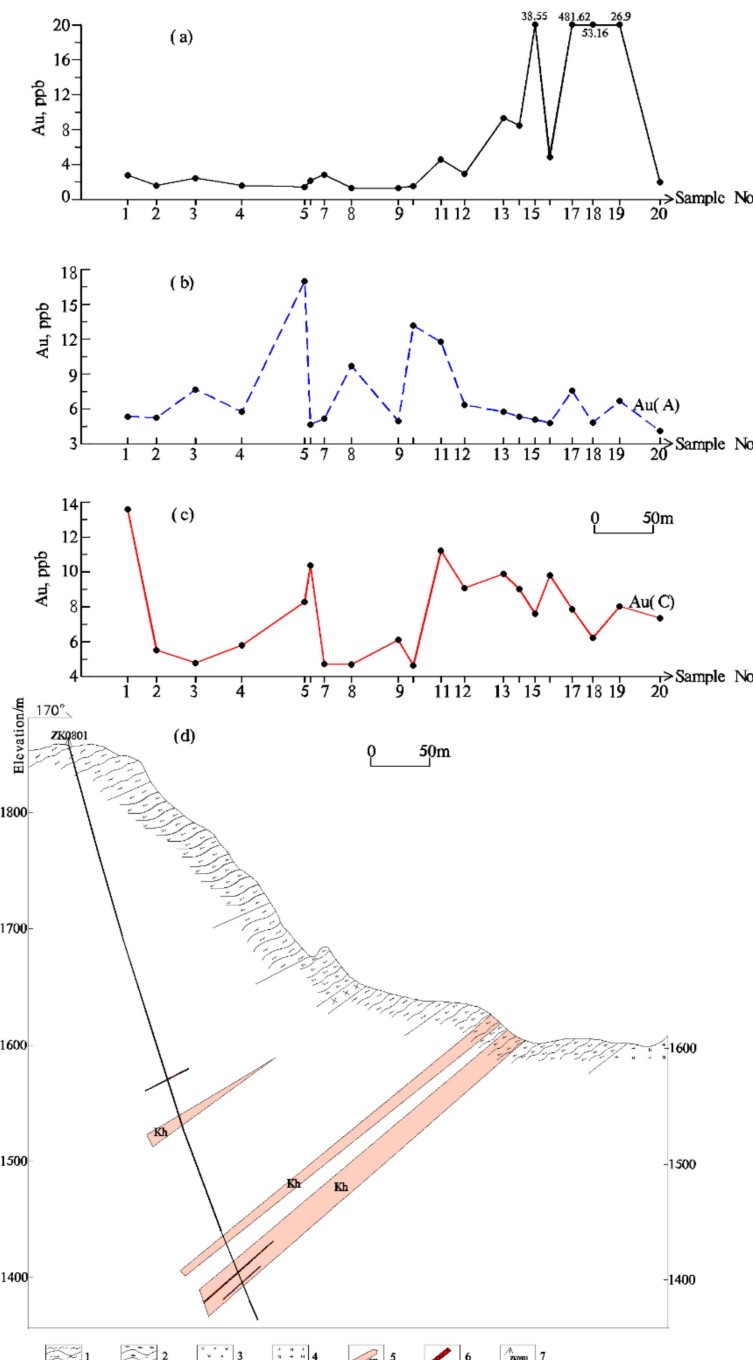

**Figure 5.** Results obtained by employing the DL-CHIM method and geochemical soil survey over Daiyinzhang gold deposit from Shanxi, China. 1—Chlorite schist; 2—Sericite schist; 3—Metamorphic diabase; 4—Plagioclase granite; 5—Gold mineralization; 6—Gold ore body; 7—Drill hole; (**a**) Soil geochemical anomalies; (**b**) Anion anomalies of Au (Anode extraction); (**c**) Cation anomalies of Au (Cathode extraction); (**d**) geological base map [35,36].

## 5. Conclusions

(1) The DL-CHIM method, characterized by simple equipment, easy operation and low cost, is therefore suitable particularly for field surveys.

(2) The DL-CHIM method enables a simultaneous extraction of anionic and cationic species of pathfinder elements, from which more information can be derived with a higher extraction efficiency.

(3)    In thick overburden areas, deep-seated concealed ore bodies are difficult to identify by conventional geochemical exploration methods, such as geochemical soil survey. However, the DL-CHIM method can detect obvious anomalies of pathfinder elements for anionic or cationic species. Additionally, anion anomalies of elements can show better results for concealed gold deposits in depth because gold may mainly exist as micro–nano particles in surface soils, which are negatively charged.

(4)    The prospecting depth of this technique is much greater than that of conventional geochemical exploration methods because the detecting objects of this technique are those mobile forms of elements and ions.

**Author Contributions:** M.K., conceptualization, methodology, data curation, writing—original draft, investigation, project administration; H.G., investigation, data curation; W.Z., investigation, funding acquisition; X.L., supervision; J.Y., writing—review and editing. All authors have read and agreed to the published version of the manuscript.

**Funding:** This work was financially supported by the National Natural Science Foundation of China (Grant No. 40743018).

**Institutional Review Board Statement:** Not applicable.

**Informed Consent Statement:** Not applicable.

**Data Availability Statement:** Not applicable.

**Conflicts of Interest:** The authors declare no conflict of interest.

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
