# Peer review of "The Improvement and Application of the Geoelectrochemical Exploration Method"

_applsci, doi:10.3390/app13042735_

Round 1

Reviewer 1 Report (Previous Reviewer 2)

Dear Authors,

I have read the work with the new suggested changes, the work can now be published.

Author Response

Thank you for your comments and suggestions. I will try my best to improve them.

Reviewer 2 Report (New Reviewer)

CHIM is an important and effective method for concealed ore exploration. The authors give a brief introduction of halo-forming mechanism and the improved equipment called Dipole CHIM. Moreover, three case studies show the importance of anode extraction for the buried gold exploration. However there are some parts in the manuscript could be improved.

1) There are many small errors in English language and style, which should be revised.

2) As I know, there are two types of CHIM in China. Unfortunately, I can not find the one proposed by Institute of Geophysical and Geochemical Exploration, CGS. Could you introduce the difference between them?

3) Some papers have discussed the halo-forming mechanism but not listed in this study, for example, Sun et al found micro-nano particles containing metals in collectors, Liu P.F. et al indroduced the halo-forming mechanism in a sketch map.

4) Figure 1 is not perfect. Both dissolved metals and mirco-nanoscale metal particles/minerals could be exracted by the anion and cation collectors. You could further modify Figure 1 upon new literatures.

5) If the soil survey could detect the buried ore body, you do not need conduct more complicated CHIM. You'd better delete soil line(figure 4-a). If you insist on keeping it, you should discuss the advantage of CHIM than soil survey.

6) In my opinion, the highlight of your study is the better result of anode extraction than that of cathode extraction. That is the biggest meaning the improved Dipole CHIM. However, it not emphasized in your results. Moreover, you should discuss why anode extraction show better result for concealed gold deposits in depth. Here is the tips,  gold may mainly exist as nanoparticles in surface soils, which is negetively charged, so gold can migrate to anode.

Author Response

Thank you for your comments and suggestions. I will try my best to improve them.

Round 2

Reviewer 2 Report (New Reviewer)

1)There are still some errors in English grammer. The paper can be futher improved by polishing of English.

2)Maybe you can modify the bottom part of Figure 1. Orebodies are not only dissoluted but also decomposed by  various geological and geochemical process. 

3)Moreover, the significance of the improved CHIM could be futher discussed in the modified manuscript.  You can emphasize the vital role of anode, especially for the gold deposit.

Author Response

Thanks for your comments and suggestions. I will revise and improve them as soon as possible. 

This manuscript is a resubmission of an earlier submission. The following is a list of the peer review reports and author responses from that submission.

Round 1

Reviewer 1 Report

The topic discussed in the manuscript is rather interesting and advances in the field of electrogeochemical exploration would be of great interest to the research community. Unfortunately the authors do not communicate their results in an organised, well-structured manner and the reader has a hard time to understand what is the actual focus and significance of this study. 

The manuscript is written in a format of disparate material deriving from different locations, grouped into a poorly structured document. It is not clear why four different locations are studied using different methods and why not all of the methods are not compared in all locations (or at least all methods in one location) to address the accuracy and reproducibility of the results. Additionally in the Taifu deposit the results are compared with the extracted ore grade while in the rest areas with the soil analyses thus it cannot be used in a paper where the authors claim that mainly compares the electrogeochemical methods (EGM) with the routine geochemical analyses (RGA)

The experimental design lacks of logical and comparative organisation to support the hypothesis that EGMs overperform the RGAs.  The comparison between the two exploration approaches (EGM vs RGA) is not scrutinised and quantified (eg use of appropriate metrics or statistics) and is given in unacceptably poorly structured images (lack of detailed explanation, inconsise with the results description, scale differences that do not support the discussion, lacking sub-labelling such as a,b,c). There are plenty of images, where some are potentially not necessary as there is redundancy information shown. The images are poorly structured, and hence they do not allow the reader to easily understand and interpret them. Improvements in the images would highlight inconsistencies between the highlighted efficiency of the EGM and the RGA and contradict the authors claims (please see relevant comments on the manuscript). Overall the authors do not convince the reader that the methodology was set up appropriately to address a specific research question. On the contrary, it seems that available data (not collected to address this research question) are used to compile a manuscript. It is quite concerning that authors present the methodological improvements as a result of this work (due to vague statements and lack of structure in the manuscript), while in reality these improvements are results of published (and cited) work.

The authors present well-rounded citations of the early advances of the EGM methods, however their most recent (after 2000) citations are purely self-citations (with the exception of one paper). Other publications from some of the authors (eg Liu et al 2019), include recent citations other than their own work, so it is hard to convince that this is a coincidence and not an effort to increase their personal metrics. It is true that the authors are involved in the majority of the recent publications on the EGMs, however, they could still present an objective and holistic overview of the recent studies on EGMs, by citing other relevant works, at least in their introduction (eg Xie and Wang, 2003, Tan, 2000, Sun et al. (2011), Sun et al., 2015, Wang and Ye, 2011, Wang et al., 2016, Wang et al., 2017).

While I am not a native English speaker, I do notice that for the most part, the manuscript is not easy read and language polishing would be required (after addressing the overall weaknesses of the work).

Overall, the results of this work should be re-written in an organised way, with a clear focus. Any analysis or location that is not comparable to the results presented and redundant information should be omitted. The results need rigorous and in-dept analysis (include quantitative techniques) to support the validity of the discussion provided. The strengths and limitations should be described in a clear, concise manner. The provided graphs do not show convincing advantage of the EGMs over the RGAs. Clearly the manuscript does not deliver the content that the title promises. There is NO a specified and quantified improvement in the described methodology. If there is no real novelty in the methodology (previously published and cited work) and no strong evidence that EGM methods over perform the RGAs, then the authors need to critically review: What exactly is the purpose and the significance of this study?

Lines 2-3: The title does not reflect the content of the study

Lines 28-29: Here the method is presented as novel and according to the abstract of Kang 2009 the same method is described for the Taifu, Yingezhuang and 210 gold ore.

Lines 88-90: These sentences are not understandable

Line 178: Not clear what is the condition without the battery, why it would be needed and in which areas is going to be used

Lines 183 -187: repetition of the same information

Lines 209-210: Inconsistent method terminology. Where is the Au in ppb in Table 1? The table formatting confuses the reader. What is the voltage in the 3rd row? 

Line 211: what are the units in the prospecting profile and Au anomaly area and number?

Line 221: Not clear, a graph (like the ones given below) would be faster to read and understand

Line 239: Inconsistent reporting; no ore grade given here. Also by whom it was discovered? During the prospecting-mining or from the electrogeochemical exploration?

Line 257: It seems that soil geochemistry is targeting better the higher Au amount (thicker line of the ore body). It is unclear why the authors use only the anionic species to highlight the differences with the soil geochemical data. The combination of the anionic and cationic species would give the almost identical result to the soil geochemistry presented

Line 260-261: These is not well supported from the results and could be even interpreted as false positives (no sensitivity to the amount of Au, so all prospecting areas seem of equal value)

Line 264: Yet Fig 9 shows only the anionic species

Lines 266-267: The authors contradict themselves twice here; firstly the don’t present the combination of the anionic and cationic species, and secondly the pattern of the optimal combination of both species they state, is very similar to the soil geochemical data. Thus, the claim the electrogeochemical exploration gives better result is not robustly supported here

Lines 269-270: True but the authors do not address the similarity with the soil geochemical data (!) And this raises again the question why the anionic species where preferred in the comparison

Lines 273-274: The figure is poorly constructed (as well as the previous ones). There is no a, b, c subgrouping to guide the reader. Top part (a): what is the value of 8.09 in a scale from 0 to 5? Same for b.: what does the value 9.46 mean? (I suppose the ore grade but authors must be explicit). Also is there is inconsistency with the method terminology. Which method do they show? CHIM or DL-CHIM? If the authors use the same scale for a and b parts then soil data give more pronounced anomaly for the two richer in Au sites (near Zk54 and at Zk71). Part c, what is the orange line? there is no explanation for the thickness of the red line

Line 283: which figure?

Lines 284- 286: Not true! Only one anomaly corresponds to the ore body. The majority are nearby but not on the ore body. They could be interpreted as false positives

Lines 288-289: Vague and potentially misleading statement. Since the majority of the anomalies are not located exactly above the ore body, it could also be a very negative point in the sense that drilling points could be suggested at site without gold deposit, which would in turn lead to capital loss

Line 290: FPAE or AFPE?? What is WAEF?

Lines 306-323: No method, results or discussion here! Only a superficial description of random facts about the Daiyinzhang gold deposit, which authors do not use of their (inexistent here) interpretations

Line 337: Vague not well-supported claim

Author Response

Thanks for reviewer's comments. I will revise the manuscript according to the comments as soon as possible.

Reviewer 2 Report

Dear Authors,

I have read the submitted work. the technique used is interesting and these preliminary data may be useful for future work of the same type. the results seem to make sense and so the work can be submitted. as you can see from the file I have attached, I think some figures need to be improved because they are not very clear and of course, the figures are very important to the readers. I am not a native English speaker but I think the English are quite correct and discursive, without obvious errors. based on my assessment the work can be submitted after less revision

Author Response

(The authors gave the same response as above.)

Round 2

Reviewer 1 Report

Dear authors,

Please find some comments with regards to changes in the manuscript. The main observed change is deletion of the questionable parts. There is no improvement in the text to support the questionable previous claims. Currently you have just disregarded half of your methodology, when your original purpose was to highlight the differences between the electrogeochemical methods and the routine geochemical work. If all these can simply omitted then what was the point of using them in the first place. This is a strong indication that this paper lacks focus and purpose. After the revisions, critical evaluation of the results and interpretations are missing. As a consequence there is no convincing evidence on how and why electrogeochemical methods are more efficient than geochemical analyses. The results presented and the relevant figures do not provide any strong evidence on that. Overall the revision effort has not addressed the critical questions raised and hence, have not improved the quality of this manuscript in a significant way

Revision Explanation: Reviewing: 1

Comment 1. Lines 2-3: The title does not reflect the content of the study. Author response: The content of the paper can basically reflect the title, and the author has also revised it according to the comments of the reviewer.

  • [ ] Lines 2-3: the improved DL-CHIM method is a result of a previous study and hence the title is still misleading. This study has not provided any improvement (as claimed by the title) in the electrogeochemical method they used. Authors have simply used the DL-CHIM in different deposits. They do not provide methodological advances as is the topic focus of this special issue

Comment 2. Lines 28-29: Here the method is presented as novel and according to the abstract of Kang 2009 the same method is described for the Taifu, Yingezhuang and 210 gold ore.Author response: The sentence has revised, and see “applsci-1835269-revision- amend”.

  • [ ] ****Lines 28-29: adequate amendment

Comment 3. Line 178: Not clear what is the condition without the battery, why it would be needed and in which areas is going to be used.Author response: the foam plastic adsorption extraction (FPAE) method was deleted.

  • [ ] Line 178: If different conditions are used, their purpose and application should be explicit. Deleting an unclear part of an actual procedure is not improving the paper

Comment 4. Lines 183 -187: repetition of the same information. Author response: Deleted the repetition information.

  • [ ] Lines 183 -187: Authors blindly deleted the repetition removing instead of combining the redundant information into one sentence. This superficial handling removes any info about the soil analyses. Additionally, since authors have removed all parts related to the FPAE method what are the foam samples is line 180?

Comment 5. Lines 209-210: Inconsistent method terminology. Where is the Au in ppb in Table 1? The table formatting confuses the reader. What is the voltage in the 3rd row? Author response: Deleted Lines 209-210.

  • [ ] Lines 209-210: Deleting an unclear part is not improving the paper. The supplementary file type was unreadable for cross check

Comment 6. Line 211: what are the units in the prospecting profile and Au anomaly area and number?Author response:Deleted Line 211.

  • [ ] ****Line 211: Deleting an unclear part is not improving the paper. The supplementary file type was unreadable for cross check

Comment 7. Line 221: Not clear, a graph (like the ones given below) would be faster to read and understand.Author response: Deleted Line 221.

  • [ ] ****Line 221: Deleting an unclear part is not improving the paper. The supplementary file type was unreadable for cross check

Comment 8. Line 239: Inconsistent reporting; no ore grade given here. Also by whom it was discovered? During the prospecting-mining or from the electrogeochemical exploration?Author response: Deleted Line 239 and relevant contents.

  • [ ] ****Line 239: Deleting an unclear part is not improving the paper. The supplementary file type was unreadable for cross check

Comment 9. Line 257: It seems that soil geochemistry is targeting better the higher Au amount (thicker line of the ore body). It is unclear why the authors use only the anionic species to highlight the differences with the soil geochemical data. The combination of the anionic and cationic species would give the almost identical result to the soil geochemistry presented.

Author response: This sentence has been rewritten**.**

  • [ ] ****Line 257: Its largest part was simply deleted, inadequate amendment

Comment 10. Line 260-261: These is not well supported from the results and could be even interpreted as false positives (no sensitivity to the amount of Au, so all prospecting areas seem of equal value)Author response: This sentence has been rewritten**.**

  • [ ] ****Line 260-261: the comment is not addressed, authors only added superficial descriptions. I would repeat the comment: how is it possible to show 40+ ppb in a plot with a scale 0 to 8 or 0 to 25 ppb? This could be considered as data/figure manipulation.

Comment 11. Line 264: Yet Fig 9 shows only the anionic species Author response: This sentence has been rewritten**.**

  • [ ] ****Line 264: it was not re-written, the relevant sentence is just deleted. Comment was not addressed adequately and clarity was not improved

Comment 12. Lines 266-267: The authors contradict themselves twice here; firstly the don’t present the combination of the anionic and cationic species, and secondly the pattern of the optimal combination of both species they state, is very similar to the soil geochemical data. Thus, the claim the electrogeochemical exploration gives better result is not robustly supported here.

Author response: This sentence has been rewritten**.**

  • [ ] ****Line 266-267: it was not re-written, the relevant sentence is just deleted. Comment was not addressed adequately and clarity was not improved

Comment 13. Lines 269-270: True but the authors do not address the similarity with the soil geochemical data (!) And this raises again the question why the anionic species where preferred in the comparison.Author response: The soil geochemical data has been provided.

  • [ ] ****Line 269-270: Indeed a mere description of the geochemical data is given (lines 272-275). However the comment pointed the need for a critical comparison of the results from the different methods, which was not addressed

Comment 14. Lines 273-274: The figure is poorly constructed (as well as the previous ones). There is no a, b, csubgrouping to guide the reader. Top part (a): what is the value of 8.09 in a scale from 0 to 5? Same for b.: what does the value 9.46 mean? (I suppose the ore grade but authors must be explicit). Also is there is inconsistency with the method terminology. Which method do they show? CHIM or DL-CHIM? If the authors use the same scale for a and b parts then soil data give more pronounced anomaly for the two richer in Au sites (near Zk54 and at Zk71). Part c, what is the orange line? there is no explanation for the thickness of the red line. Author response: This paragraph has been revised**.**

  • [ ] ****Lines 273-274: The figure has been partly improved but authors have not provided clarifications to all the comments. Also fig.4d is not geological base map, it’s a geological cross-section. If fig4.d data are not results of this study authors should have provided a citation. Authors claim they have improved the paragraph but the comment was for a figure. It is unclear what revisions they mean here. Clarified parts

    • a,b,c,d groupings
    • Which method do they show? → DL-CHIM

    Not clarified parts:

    • Fig 4.a what is the value of 8.09 in a scale from 0 to 5? → now is 82.87, 45.06 on a scale from 0-45 ppb
    • Same for b.: what does the value 9.46 mean? → now is 41.05 in a scale from 0-8 ppb
    • If the authors use the same scale for a and b parts… → important comment which not addressed at all
    • Part c, what is the orange line? → not addressed (it’s now fig. 4d)

Comment 15. Line 283: which figure? Author response: That is Fig. 5, which is the newly replaced Figure.

  • [ ] ****Line 283: clear amedment

Comment 16. Lines 284- 286: Not true! Only one anomaly corresponds to the ore body. The majority are nearby but not on the ore body. They could be interpreted as false positives.Author response: This sentence has been rewritten**.**

  • [ ] ****Lines 284- 286: inadequate amendment. Authors now mention there is only one anomaly for the cationic species (in lines 288-289), however they do not describe clearly the the anomalies right next to the ore body, neither how this compares to the geochemical data

Comment 17. Lines 288-289: Vague and potentially misleading statement. Since the majority of the anomalies are not located exactly above the ore body, it could also be a very negative point in the sense that drilling points could be suggested at site without gold deposit, which would in turn lead to capital loss. Author response: This sentence has been revised

  • [ ] ****Line 288-289: inadequate amendment. same comment as above

Comment 18. Line 290: FPAE or AFPE?? What is WAEF? Author response: This sentence has been deleted.

  • [ ] ****Line 290: the whole method and its results are deleted. Authors did not try to mprove for clarity, instead the omitted completely the unclear parts

Comment 19. Lines 306-323: No method, results or discussion here! Only a superficial description of random facts about the Daiyinzhang gold deposit, which authors do not use of their (inexistent here) interpretations.Author response: The method has described in the 3.2.2 and 4.2, and analytical results of this work are tabulated in Table 3, and relevant references are added.

  • [ ] ****Lines 306-323: Authors response on the method could be adequate as now they only discuss one method, comparing to the previous manuscript version. However the comment is not addressed, no interpretations are given only a superficial description about the Daiyinzhang gold deposit

Comment 20. Line 337: Vague not well-supported claim. Author response: This sentence has been rewritten**.**

  • [ ] ****Line 337: the sentence is rewritten without practically changing the meaning which remains vague and unsupported

Reviewing: 2 Author response: According the reviewer's suggestion, Fig.1 and Fig.2 have been

redrawn to make the text clearer, and Fig.3 has been enlarged.

  • [ ] ****Indeed this is modified accordingly

Author response: According the reviewer's suggestion, I used a, b, c and d letters to distinguish the sub-figures in Fig.4, Fig.5 and Fig.6.

  • [ ] ****Indeed this is modified accordingly

Author Response

Thanks for your suggestions, and I will try my best to revise the manuscript again.

Ming

Round 3

Reviewer 1 Report

Dear authors,

In the revised version I see in the system, I did not find any changes related to the comments addressed in the second review report.
The only change the addition of 2 identical sentences where the only change is the sample numbers.
Since my previous comments were not addressed I do not have additional comments.

Author Response

Thank  you very much for your advice.